# Effects of Dietary *Gracilaria lichenoides* and *Bacillus amyloliquefaciens* on Growth Performance, Antioxidant Capacity, and Intestinal Health of *Penaeus monodon*

**DOI:** 10.3390/biology13040252

**Published:** 2024-04-09

**Authors:** Jialin Tian, Yun Wang, Jianhua Huang, Hailiang Yan, Yafei Duan, Jun Wang, Chuangpeng Zhou, Zhong Huang

**Affiliations:** 1College of Fisheries and Life Science, Shanghai Ocean University, Shanghai 201306, China; jltian2023@163.com (J.T.);hailiangyan813@163.com (H.Y.); 2Key Laboratory of Aquatic Product Processing, Ministry of Agriculture and Rural Affairs, South China Sea Fisheries Research Institute, Chinese Academy of Fishery Sciences, Guangzhou 510300, China; hjh210440@sina.com.cn (J.H.); duanyafei89@163.com (Y.D.); junnywang@163.com (J.W.);; 3Key Laboratory of Efficient Utilization and Processing of Marine Fishery Resources of Hainan Province, Hainan Engineering Research Center of Deep-Sea Aquaculture and Processing, Sanya Tropical Fisheries Research Institute, Sanya 572018, China; 4Shenzhen Base of South China Sea Fisheries Research Institute, Chinese Academy of Fishery Sciences, Shenzhen 518121, China; huangzhongnhs@163.com

**Keywords:** *Penaeus monodon*, *Gracilaria lichenoides*, *Bacillus amyloliquefaciens*, growth, antioxidant capacity, intestine microbiota, lipid metabolism

## Abstract

**Simple Summary:**

Seaweeds and probiotics are commonly utilized as additives in the diet of aquatic animals. However, limited information exists regarding the effects of *Gracilaria lichenoides* and *Bacillus amyloliquefaciens*, either alone or in combination, on *Penaeus monodon*. In our study, we examined the impact of dietary supplementation with *G. lichenoides* and *B. amyloliquefaciens*, both individually and in combination, on the growth performance, antioxidant capacity, and intestinal function of *P. monodon*. Our findings indicate a beneficial influence of dietary *G. lichenoides* and *B*. *amyloliquefaciens* on *P. monodon*. These findings offer a theoretical foundation for the judicious utilization of *G. lichenoides* and *B. amyloliquefaciens* in the cultivation of black tiger shrimp, as well as for future investigations into their metabolic mechanisms.

**Abstract:**

This research sought to assess the effects of dietary supplements with *Gracilaria lichenoides* and *Bacillus amyloliquefaciens*, either individually or combined, on the growth performance, antioxidant capacity, and intestinal function of *Penaeus monodon*. A total of 840 shrimps were randomly assigned to 28 tanks with an average initial weight of (1.04 ± 0.03) g (30 shrimp per tank) with 7 different treatment groups and 4 replicates per treatment. The control treatment (C) consisted of a basal diet; in contrast, the experimental groups were complement with varying levels of *G. lichenoides* (3% or 8%), either alone (S3 and S8) or in combination with *B.amyloliquefaciens* at different concentrations (3% *G. lichenoides* and 10^9^ CFU/g—S3B9; 8% *G. lichenoides* and 10^11^ CFU/g *B. amyloliquefaciens*—S8B11; 10^9^ CFU/g *B. amyloliquefaciens*—S9; 10^11^ CFU/g *B. amyloliquefaciens*—B11). The results indicated that the maximum values of final body weight (FBW) (10.49 ± 0.90) g, weight gain rate (WGR) (908.94 ± 33.58) g, and specific growth rate (SGR) (4.20 ± 0.06) g were perceived in the 3% *G. lichenoide* diet treatment, and compared with the control group, the difference was significant (*p* < 0.05). The whole-body lipid content of shrimp in the B9 group was significantly higher than that in the B11 group (*p* < 0.05), but no significant difference was observed when compared with shrimp fed other diets (*p* > 0.05). The ash content of shrimp in the B9 group was found to be significantly higher than that in the S3B9 group (*p* < 0.05). Furthermore, the lipase activity in the stomach and intestines of the experimental groups exhibited a statistically significantly increase compared to the control (*p* < 0.05). In comparison to the control group, the hepatopancreas of the S3 group exhibited a significant increase in the activities of glutathione peroxidase (GSH-Px), superoxide dismutase (SOD), and antioxidant genes [*SOD*, catalase (*CAT*), *GSH-Px*, thioredoxin (*Trx*), *Hippo*, and NF-E2-related factor 2 (*Nrf2*)] expression levels (*p* < 0.05). Additionally, the activities of total antioxidant capacity (T-AOC), SOD, peroxidase (POD), and antioxidant genes (*CAT*, *GSH-Px*, *Trx*, and *Hippo*) in the S3B9 treatment of hepatopancreas showed significant improvement (*p* < 0.05). The inclusion of dietary *G. lichenoides* and *B. amyloliquefaciens* resulted in enhanced relative expression of intestinal lipid metabolism genes (fatty acid synthetase (*FAS*), lipophorin receptor (*LR*), fatty acid transport protein 1 (*FATP*1)) and suppressed the expression of the long-chain fatty acid-CoA ligase 4 (*LCL*4) gene. Analysis of microbiota sequencing indicated improvements in composition and structure, with notable increases in Firmicutes at the phylum level and *Vibrio* at the genus level in the S3 group, as well as an increase in *Tenericutes* at the genus level in the S8B11 group. Overall, the inclusion of dietary *G. lichenoides* and *B*. *amyloliquefaciens* positively impacted the growth, antioxidant capacity, and microbial composition of shrimp, with particular enhancement observed in shrimp fed a supplementary 3% *G. lichenoides* diet.

## 1. Introduction

*Penaeus mondon*, also known as black tiger shrimp, is the second largest shrimp species in China, with an annual production of over 100,000 tons [1]. *P. monodon*, as a traditional cultured shrimp, has the characteristics of strong adaptability, fast growth, strong disease resistance, delicious flavor, high nutritional value, and resistance to transportation [2]. However, diseases are still the most important constraints affecting the development of shrimp culture. Also, water quality deterioration and serious pollution leads to an increase of the pathogenic microorganisms. Antibiotic use in aquaculture is often applied to shrimp farms to overcome disease and stress problems. In response to these problems, antibiotics are often used in shrimp production and medicine. To combat the adverse effects of antibiotics, researchers have been looking for alternatives. Additionally, consumers demand growing levels of quality and safety for shrimp farming, as well as an absence of antibiotics and pollutants. As a result, shrimp farming strategies must accelerate the search for safe dietary supplements and additives that enhance their health, vital activities, and environmental resilience. Antioxidants and microbes added to feed have been considered a non-antibiotic approach that can improve health and performance [3]. 

There are several efforts underway to develop products or methods for treating or preventing shrimp farm stress [4]. Macroalgae (seaweeds) are rich in bioactive compounds that can potentially be exploited as functional ingredients in feed additives or supplements for animal health applications [5]. *Gracilaria lichenoides*, belonging to the division Rhodophyta, is an important aquaculture species with good bioremediation potential due to their fast growth rate, high nutrient uptake capacity, and high temperature tolerance [6,7]. A significant portion of the lipid content of seaweeds comes from polyunsaturated fatty acids (PUFAs), especially those of tropical origin [8]. Aside from its polysaccharides and dietary fibers, the seaweeds is known to contain incomparable amounts of minerals, vitamins, and polyphenols [9]. The utilization of seaweed as a dietary supplement for fish and shrimp at low concentrations (2.5–10%) has demonstrated beneficial outcomes, such as enhanced growth performance, increased feed utilization [10,11,12,13], and improved immune and stress responses [14,15]. Supplementation of *Sciaenops ocellatus* feed with *Isochrysis galbana* extract, resulted in significantly higher survival rates [16]. Feeding *Oncorhynchus mykiss Gracilaria pygmaea* polysaccharide led to significantly higher final average weight compared to the control group, with feed intake showing a tendency to increase with higher levels of supplementation [17]. Incorporating *Gracilaria gracilis* into the diet of *Sparus aurata* enhanced weight gain rate and feed conversion ratio [18]. The incorporation of *Eucheuma denticulatum* into the diet of *Paralichthys olivaceus* resulted in notable enhancement in the growth rate and feed efficiency of juvenile fish, with a concurrent increase in the accumulation of dorsal muscle omega-3 polyunsaturated fatty acid in experimental groups [19]. Additionally, feeding *Gracilaria persica* to *Acipenser persicus* led to elevated activities of superoxide dismutase (SOD) and catalase (CAT) [20]. Furthermore, supplementation of *Sargassum aquifolium* in the feed resulted in heightened activities of CAT, SOD, and glutathione (GSH), along with a reduction in malondialdehyde (MDA) content. Notable, *SOD* and *CAT* were significantly upregulated in *Oreochromis niloticus* [21]. *Sargassum horneri* was found to enhance hepatic total antioxidant capacity (T-AOC), SOD, and CAT activities in *Acanthopagrus schlegelii*, with a notable upregulation of *CAT* expression [22]. Similarly, the dietary supplementation of sulphated polysaccharide resulted in a significant increase in hepatic T-AOC, CAT, and SOD activities, as well as marked upregulated *Cu^2+^/ZnSOD*, *CAT*, and glutathione peroxidase (*GSH-Px*) antioxidant gene expression in *Siganus canaliculatus* [23]. In contrast, some researchers found that seaweeds and their extracts induced several morphological changes in fish guts, inhibiting digestion and nutrient absorption [24,25,26]. *Gracilaria lemaneiformis* increased the length and evenness of intestinal microvilli in *L. vannamei* [27]. Niu et al. [14] found that the addition of *Undaria pinnatifida* to the feed significantly increased the height and width of folds in shrimp, as well as the lipase activity. Extracts of *Amphiroa Fragilissima* promoted digestive enzyme (protease and amylase) activities in *P*. *monodon* [28].

Probiotics have been defined as “live microorganisms” which work by rebuilding the gastrointestinal trace (GIT)’s native microflora in response to ingestion of food or water [29]. *Bacillus amyloliquefaciens*, known as probiotic, which is closely related to *Bacillus subtilits*, can improve the intestinal microbial environment, facilitate aquatic animal growth, improve enzyme activity, and boost nonspecific immune responses [30,31]. In our earlier research, a specific probiotic *B. amyloliquefaciens* broke away the healthy *L. vannamei* intestine, which is an endogenous strain that has strong antagonistic effects against seven species of *Vibrio spp*. pathogens in aquatic animals. Besides, it can also improve the non-specific immune function, relieve oxidative stress, and strengthen the ability of shrimp to resist pathogenic bacterial infections. Similar results were also found when a strain of *B. amyloliquefaciens* was isolated from the intestine of healthy *P. clarkia*, exhibiting strong antimicrobial activity against different pathogens. *B. amyloliquefaciens* as a feed additive could improve the intestinal digestive enzyme activity, the innate immune enzyme activities, and the antiviral resistance of *P. clarkia* [32]. *Bacillus* also has beneficial effects on the microbial community and the development of *L. vannamei.* Wang et al. explained the effects of *Bacillus* (10^8^~10^9^ CFU/g) on the survival rate and molting of *L. vannamei* larvae [33]. *B. amyloliquefaciens* significantly improved the WG, SGR, and FCR of *O. niloticus* [34]. *B. amyloliquefaciens* LSG2-8 added to *Rhynchocypris lagowskii* feed significantly improved FBM, WGR, SGR, and FCR growth indices [35]. *L. vannamei* phenol oxidase activity, phagocytic activity, and scavenging efficiency were significantly increased by feeding higher concentrations of *Bacillus subtilis* E20 [36]. *B. amyloliquefaciens* R8 caused a significant increase in decreased mRNA expression of oxidative stress-related genes (*SOD*, *Gpx*, and *Hsp70*) in zebrafish liver [37].

The dietary role of *G. lichenoides* and *B. amyloliquefaciens*, either alone or in combination, on digestive and metabolic capacity regulatory in *P. monodon* remains uncertain. The potential benefits of incorporating *G. lichenoides* and *B. amyloliquefaciens* into shrimp diets were previously unidentified. Therefore, this study examined the impact of *G. lichenoides* and *B. amyloliquefaciens*, either individually or in combination, on the growth, antioxidant capacity, and intestinal health of *P. mondon*. The aim was to identify the most effective dosage and application method of *G. lichenoides* and *B. amyloliquefaciens* in the diet of *P. mondon*, optimize feed formulation, and enhance feed utilization efficiency and antioxidant stress response in *P. mondon*. Further, the effects of *G. lichenoides* and *B. amyloliquefaciens* on the intestinal microbiota and overall intestinal health of *P. mondon* was explored, with the goal of establishing a theoretical framework for research in black tiger shrimp aquaculture and elucidating their metabolic mechanisms.

## 2. Materials and Methods

### 2.1. Ethical Statement

This animal study was conducted under animal use protocol number NHDF2023-12 and was approved by the Laboratory Animal Welfare and Ethics Committee of the South China Sea Fisheries Research Institute, Chinese Academy of Fishery Sciences.

### 2.2. Experimental Diets Preparation and Nutritional Composition

Basic feed ingredients were purchased from Shandong Wuzhoufeng Agricultural Science and Technology Co., Ltd (Heze, China). *G. lichenoide* was purchased from the breeding farm of Xigao Village, Nanri Town, Xiuyu District, Putian City, Fujian Province. *B. amyloliquefacients* was supplied by our laboratory, isolated from the intestinal tract of healthy *L. vannamei* (Patent No.: ZL201410319443.7). Seven experimental diets were prepared based on the basal diet of black tiger shrimp. The basal diet served as the control, while the other diets included the addition of 3% *G. lichenoide* as S3, the addition of 3% *G. lichenoide* and 10^9^ CFU/g *B. amyloliquefacients* as S3B9, the addition of 10^9^ CFU/g *B.amyloliquefacients* as S9, the addition of 8% *G. lichenoide* as S8, the addition of 8% *G. lichenoide* and 10^11^ CFU/g *B. amyloliquefacients* as S8B11, and the addition of 10^11^ CFU/g B. amyloliquefacients as B11. The feed formulation and composition are analyzed in Table 1. Feed preparation was done according to the method of feed preparation in our laboratory [38]. The feed was made into pellets with diameters of 1.0 mm and 1.5 mm and were heated in an oven at 90 °C for 90 min, and then air-dried to about 10% moisture and stored in a −20 °C freezer. Moisture, crude protein, crude lipid, and ash of the experimental diets were determined using standard methods [39]. 

### 2.3. Shrimp and Culture System

Juvenile black tiger shrimp were obtained from the Shenzhen Experimental Base of the South China Sea Fisheries Research Institute of the Chinese Academy of Fishery Sciences. Shrimp were domesticated with commercial feed for one week before the beginning of the experiment. Shrimp with an initial average weight of (1.04 ± 0.03) g were randomly assigned to 28 fiberglass tanks (500 L, 0.5 m^2^ round bottom area, 30 shrimp per tank) in 7 groups of 4 tanks each. During the feeding period, shrimp were fed at 8:00, 17:00, and 22:00 every day, and the daily feeding regimen consisted of providing shrimp with an amount equivalent to 5% of their body weight, closing the screen after each feeding to prevent escape, and adjusting the feeding quantity promptly based on shrimp behavior and prevailing weather conditions. After 1 h of feeding, the remaining shrimp feed in the bait dish was collected, rinsed 2 to 3 times with water, dried at 60 °C, and weighed. The numbers of shrimp deaths and body weight of each tank were recorded. Oxygen was connected to the oxygen supply system by a hose air stone, which was filled day and night. The breeding water was filtered seawater. The rearing water was siphoned off the bottom every day, and 1/3 of the rearing water was changed on a weekly basis. During the experiment, water parameters were maintained as follows: temperature 29.7 ± 0.9 °C, pH 8.0 ± 0.1, salinity 30, dissolved oxygen more than 6.0 mg/L. The feeding trial lasted for 56 days.

### 2.4. Sampling and Preservation

At the end of the feeding trial, shrimp were fasted for 24 h, and then weighed to determine growth performance. Five shrimp from each tank were randomly selected for whole-body composition analysis. Additionally, five shrimp from each tank were sampled for analysis of enzyme activity and gene expression in the hepatopancreas and intestines. The intestines from two shrimp per tank were extracted at 56 d and stored in Davidson’s solution for 24 h. The intestines from five shrimp per tank were extracted and quickly frozen in liquid nitrogen, followed by storage at −80 °C. To reduce the impact of interindividual differences, intestines from each treatment were mixed for the analysis of the intestinal microbiota.

### 2.5. Measurement and Analysis

#### 2.5.1. Growth Performance

The growth parameters were calculated according to the following formulas:Weight gain rate (WGR, %) = 100 × [FBW (g) − IBW (g)]/IBW (g)
Specific growth rate (SGR, %) = 100 × [ln(FBW) − ln(IBW)]/56 days
Feed conversion ratio (FCR) = 100 × dry feed intake (g)/body weight gain (g)
Survival rate (SR, %) = 100 × final living numbers (shrimp)/initial numbers (shrimp)
Note: initial weight (IBW), final body weight (FBW)

#### 2.5.2. Activity of Physiological and Biochemical Indexes

The hepatopancreas samples were homogenized in sterilized PBS buffer (0.86%, pH = 7.4; 1: 9, *w*/*v*) by a handheld homogenizer; then, the samples were centrifuged for 10 min (3000 r/min, 4 °C) and the supernatant was fractionated [40]. Hepatopancreas T-AOC, SOD, CAT, GSH-Px, peroxidase (POD) activities, and MDA and total protein contents were determined according to the kit (Nanjing Jiancheng Bioengineering Institute, Nanjing, China). The lipase activity in the stomach and intestine were also analyzed with commercial test kits (Nanjing Jiancheng Bioengineering Institute, Nanjing, China).

#### 2.5.3. RNA Extraction and qPCR Measurement

RNA from hepatopancreas and intestine was extracted using TRIzol reagent (QIAGEN Cat: no. 74104, Guangzhou, China) according to the manufacturer’s instructions, and DNase I (QIAGEN Cat: no. 79254) was used to remove contaminated DNA. The primer sequences for the target gene and the internal reference gene *EF*-1α for real-time PCR are shown in Table 2 [41]. All primer pairs for real-time PCR were synthesized by Sangon Biotech (Shanghai) Co., Ltd. (Shanghai, China). The mRNA expression levels of the target genes were quantified using the fluorescence quantitative PCR method described in a previous report. Each sample was tested three times using real-time PCR (*n* = 4). The gene expressions were quantified using the 2^-ΔΔct^ calculation method with an internal reference gene [42]. 

#### 2.5.4. HE Stain of the Intestinal Tissue

Two shrimp from each tank had their intestines sampled and preserved in Davidson’s solution and ethanol. The tissues were then rinsed, dehydrated, and prepared for staining and photography. Eight samples were randomly selected for measuring the height of the intestinal epithelial cell and intestinal folds height of shrimp.

#### 2.5.5. Intestinal Microbiome Analysis

Genomic DNA was extracted from intestinal microbial samples using the OMEGA Soil DNA Kit (M5635-02) (Omega Bio-Tek, Norcross, GA, USA) and stored at −20 °C. The bacterial 16S rRNA genes V3-V4 region was amplified using primers 338F (5′-ACTCCTACGGGAGGCAGCA-3′) and 806R (5′-GGACTACHVGGGTWTCTAAT-3′), and the resulting PCR products were purified and combined in equal amounts. After quantifying the individuals, the amplicons were pooled and sequenced using the Illumina NovaSeq platform with the NovaSeq 6000 SP Reagent Kit (500 cycles) at Suzhou PANOMIX Biomedical Tech Co., Ltd (Suzhou, China). Microbiota sequencing and analysis were conducted at the same facility. Microbiota sequencing and subsequent analyses were performed at Suzhou Panomix Biomedical Tech Co., Ltd (Suzhou, China). 

### 2.6. Statistical Analysis

The mean ± SD of the data was presented. Statistical analysis was conducted using SPSS 26.0 software (IBM Corporation, Somers, NY, USA) for Windows. Prior to performing ANOVA, the normality and homogeneity of the data were tested using the Kolmogorov-Smirnov test and Levene’s test, respectively. Once the data passed the tests, a one-way ANOVA was performed. If there were notable differences, the group means were compared using Duncan’s multiple-range test, with a significance level of *p* < 0.05. 

## 3. Results

### 3.1. Growth Performance

Growth performance of *P. monodon* is presented in Table 3. There were no significant differences in IBW, FCR, and SR among the groups (*p* > 0.05). However, shrimp fed diets with 3% *G. lichenoide* had significantly higher FBW, WGR, and SGR compared to the control group (*p* < 0.05). 

### 3.2. Whole-Body Composition

Table 4 shows the composition of shrimps’ whole-body composition on different diets. There were no significant differences in moisture and protein content among the diet treatments (*p* > 0.05). The lipid content of shrimp in the B9 group was higher than that of shrimp in the B11 group (*p* < 0.05), but not significantly different from shrimp fed other diets (*p* > 0.05). The ash content of shrimp in the B9 group was higher than that of shrimp in the S3B9 group (*p* < 0.05). 

### 3.3. Change of Antioxidant Enzyme Activities in the Hepatopancreas

Activities of T-AOC, SOD, GSH-Px, CAT, POD, and MDA content in the hepatopancreas of shrimp for all diet treatments are shown in Figure 1. T-AOC and SOD activities were significantly higher in the S3 and S3B9 groups compared to the other groups (*p* < 0.05). CAT activity was significantly higher in the S3B9 group compared to the B9, S8B11, and B11 groups (*p* < 0.05), but not significantly different from the control, S3, and S8 groups (*p* > 0.05). The GSH-Px activity in shrimp from the S3 group was significantly higher than in the control and B9 groups (*p* < 0.05), but there was no significant difference with the other groups (*p* > 0.05). The POD activity in the S3B9 group was significantly higher than in the control, B9, S8B11, and B11 groups (*p* < 0.05). There was no significant difference in GSH-Px activity among the S3, S3B9, and S8 groups (*p* > 0.05). The MDA content in shrimp from S3 group was significantly higher than in the control, B9, S8B11, and B11 groups (*p* < 0.05). The antioxidant capacity in hepatopancreatic tissues of *P. monodon* was significantly increased in the 3% *G. lichenoide* group (S3) and the 3% *G. lichenoides* and 10^9^ CFU/g *B. amyloliquefaciens* (S3B9).

### 3.4. The Relative Expression of Antioxidant Genes in the Hepatopancreas of Shrimp

The relative expression of antioxidant genes in the hepatopancreas are showed in Figure 2. The relative expression level of *mtMnSOD* in the S3 group was significantly higher than that of shrimp in other groups (*p* < 0.05). However, the relative expression of SOD in the S8, S8B11, and B11 groups was significantly lower than that of shrimp in the control group (*p* < 0.05). Regardless of the supplementation of *G. lichenoides* and *B. amyloliquefaciens*, either individually or in combination in the diet, the relative expressions of *CAT* and *GSH-Px* in the hepatopancreas were significantly increased compared to the control group (*p* < 0.05). Furthermore, the *CAT* and *GSH-Px* genes had the highest expression in the S3B9 and S3 groups, respectively. The expression of *Trx* in shrimp increased significantly in the S3, S3B9, and S8B11 groups compared to the control (*p* < 0.05), with the highest levels observed in the S3 group. The expression of *Hippo* in shrimp was significantly higher in the S3 and S3B9 groups compared to the other groups (*p* < 0.05). The *Nrf2* gene relative expression in shrimp was significantly higher in the S3, S8, and S8B11 groups compared to other groups (*p* < 0.05), with the highest levels observed in the S8B11 group. The results indicate a significant increase in antioxidant gene expression in the hepatopancreas of shrimp between the 3% *G. lichenoide* group and the 3% *G. lichenoide* + 10^9^ CFU/g *B. amyloliquefaciens* group.

### 3.5. Lipase Activity

The results of stomach and intestinal lipase activity are shown in Figure 3. The lipase activity of intestine and stomach in the experimental groups were significantly higher than those of the control group (*p* < 0.05). Results showed that supplementing the diet with the *G. lichenoide* and *B. amyloliquefacients* improved the stomach and intestinal lipase activity of *P. monodon*.

### 3.6. Intestinal Tissue Structure

After staining with HE dye, the intestinal tissue of *P. monodon* showed a higher number of epithelium and neatly arranged brush borders in the experimental group compared to the control group (Figure 4). Vacuoles were almost absent and nuclei in S3 and S8 groups were closely arranged. The height of intestine epithelium in the S3 group was significantly higher than in the S8B11 group (*p* < 0.05). The folds depth in the S3B9, B9, and B11 groups was also significantly higher than in other groups (*p* < 0.05). Overall, supplementing the diet with *B. amyloliquefaciens* (10^9^ CFU/g) improved the intestinal tissue structure of *P. monodon*. 

### 3.7. Intestinal Lipid Metabolism Genes Expression

The expression of lipid metabolism genes in the intestine is illustrated in Figure 5. The relative expression levels of fatty acid *FAS* and *LR* genes in the shrimp intestine was significantly higher in the groups supplemented with *G. lichenoides* and *B. amyloliquefaciens*, either individually or in combination, compared to the control group (*p* < 0.05). The B9 and S8B11 groups had the highest expression levels of *FAS* and *LR* genes. The relative expression levels of intestinal fatty acid transport protein 1 (*FATP1*) were significantly higher in the B9 and S8B11 groups compared to other groups (*p* < 0.05), while the expression of *FATP1* was lower in the S3, S3B9, S8, and B11 groups compared to the control group (*p* < 0.05). The expression of long-chain fatty acid-CoA ligase 4 (*LCL4*) was lower in all experimental groups with *G. lichenoides* and *B. amyloliquefaciens* supplementation compared to control (*p* < 0.05). The results reveal that adding *B.amyloliquefaciens* (10^9^ CFU/g) and 8% *G. lichenoide* + *B. amyloliquefaciens* (10^11^ CFU/g) had a significant impact on the expression of genes involved in shrimp intestinal lipid metabolism.

### 3.8. Intestinal Microbiota Changes

#### 3.8.1. Richness and Diversity

A total of 2,535,118 high-quality sequences were obtained from the intestine microbiota of *P. monodon*. The sequence length was from 159 to 437 bp. The core OUT (operational taxonomic unit) accounted for 394 (Figure 6). The microbiota coverage of each group was all close to 99%, which suggests that the sequencing depth of this intestinal microbiota analysis was sufficient. Thus, the experimental results were representative. The bacterial richness analysis showed that Chao1 index was increased in the S8, B9 and B11 groups compared with the control group, while there was no significant difference among the groups (*p* > 0.05) (Table 5).

#### 3.8.2. Analysis of Microbial Community Composition

A total of 43 bacterial phyla were identified in the seven groups. Of the dominant phylum, the relative abundances of Tenericutes in the S8B11 groups were increased compared to the control, S3, S3B9, and S8 groups (*p* < 0.05), yet the relative abundance of Firmicutes in the S3 group was significantly higher than that in the S3B9, B9, S8, and B11 groups (*p* < 0.05) (Figure 7A). As shown in Figure 7B, at the genus level, the relative abundance of *Vibrio* in the S3 group were significantly higher than that in the S3B9 (*p* < 0.05) and had no significant difference with other groups (*p* > 0.05). Moreover, the relative abundance of *Shimia* in the B9 group was significantly higher than that of the control, S3, and S8 groups (*p* < 0.05). However, the relative abundances of *Lactobacillus* in the S3B9, B9, and S8 groups were significantly lower than that of the control group (*p* < 0.05). 

The linear discriminant analysis (LDA) effect size (Lefse) package was used to determine the differential abundances of microbiota taxa among the groups. There are 23 bacteria taxa distinguishing among the experimental groups, except the S8 and B9 groups. The relative abundance of 5, 8, 1, 5, and 4 biomarkers (LDA score > 2.0) was higher in the control, S3, B11, S3B9, and S8B11 groups, respectively. LDA scores of Lefse showed that 8 taxa were increased in the S3 group and there were two groups only enriched at genus level, including *Altomonas* and *Clostridium* (Figure 7A). From the results of Lefse cladogram (Figure 8B), 1 phylum, 1 class, 1 order, 4 families, and 2 genera were enriched in the S3, including Firmicutes (from phylum to genus), while 1 phylum, 2 class, and 1 order were enriched in the S8B11 group, including Flavobacteriia (from class to order). At the order level, Acidobacteriales is the possible biomarker for the S3B9 group, while *Lactrobacillales* was found at the control group. However, no suitable biomarker was found in the S8 and B9 groups. 

## 4. Discussion

The findings of this study demonstrate that the inclusion of a small quantity of seaweed power in the diet resulted in significantly higher FBW, WGR, and SGR in the S3 group compared to the other groups, suggesting a positive impact on the growth of prawns. Niu et al. [14] conducted an experiment in which various concentrations of wakame were administered to spotted prawns, resulting in a significant increase in WG and SGR at the 2% concentration level. Similarly, Jumah et al. [43] supplemented prawn feeds with *k-carrageenan*, resulting in a significant increase in the FBW, WG, and SGR in the experimental group. Anil et al. [44] demonstrated that the inclusion of *Solieriaceae* in the diet of shrimp resulted in increased average daily gain and total weight, suggesting that seaweed have the potential to enhance the growth performance of shrimp. The beneficial effects of seaweeds on shrimp growth can be attributed to their rich content of vitamins, mineral contents, polyunsaturated fatty acids, and amino acids [45]. The active compounds have the ability to modulate the composition of bacterial flora, leading to enhanced intestinal function that facilitates the absorption of carbohydrates and fats in the diet [45]. The growth-promoting impact of seaweeds in the diet can be linked to the digestion and absorption of key nutrients in shrimp feed, which is facilitated by the release of hydrolytic enzymes from shrimps’ digestive glands [46]. The results indicate that the S3 group exhibited significantly higher lipase activity compared to the control group, implying that the inclusion of a small quantity of *G. lichenoides* may enhance feed digestion and absorption in shrimp, leading to improved growth performance.

The process of lipid metabolism plays a crucial role in the growth and development of shrimp. Lipid transport, lipolysis, and lipid synthesis collectively govern lipid metabolism [47]. The synthesis of lipids, specifically triglycerides, necessitates the catalytic activity of FAS within the fatty acid synthesis pathway, thereby exerting a significant influence on lipid synthesis [47,48]. Furthermore, a decrease in adipogenesis is indicated by the downregulation of *FAS* [49]. *FATPs* are essential for lipid transport and synthesis, as they exhibit lipo acyl-coenzyme-A synthetase activity and facilitate the transmembrane transport of long-chain fatty acid, ultimately contributing to fat deposition [50]. *LR*s are integral membrane proteins that facilitate the binding of lipoproteins and their corresponding ligands, thereby facilitating the cellular uptake and metabolism of lipoproteins [51]. *LCL*, as a superfamily of membrane proteins, catalyze the activation of fatty acids and the formation of acyl-cofactor thioesters, serving as crucial components in both anabolic and catabolic pathways involving fatty acids [52]. In conclusion, upregulation of the genes FAS, *FATP*, *LR*, and *LCL* has been found to be advantageous for lipid synthesis. The findings of this research indicate that the expression of genes related to fat synthesis, specifically *FAS*, *FATP1*, and *LR*, was increased in the intestines of shrimp in the B9 group compared to the B11 group. This difference in gene expression may account for the significantly higher lipid content observed in the whole body of shrimp of the B9 group compared to the B11 group, suggesting a potential dose-response relationship between *B. amyloliquefaciens* and lipid metabolism. Similar results indicated a rise in muscle and hepatopancreatic fat content in *L. vannamei* following the addition of *B. subtilis* in diet [53], as well as a notable increase in the expression levels of genes associated with lipid synthesis in grass carp after the supplementation *B. subtilis* H2 in diets [54]. 

The hepatopancreas plays a crucial role as an antioxidant tissue organ in shrimp, as animals generate significant quantities of reactive oxygen species (ROS) under stress conditions, leading to oxidative damage to proteins, lipids, and DNA [49,55]. The measurement of MDA content in tissues and cells serves as an indicator of the level of oxidative damage [56,57]. Following oxidative damage, the antioxidant enzymes within the biological antioxidant enzyme system (SOD, GSH-PX, CAT, POD) work to eliminate excess ROS and safeguard the organism from harm [58]. The magnitude of T-AOC is indicative of the organism’s ability to compensate for external stimuli through its antioxidant enzyme system and non-enzymatic system, as well as its metabolic state of free radicals [59]. This measurement can serve as a reflection of the organism’s overall antioxidant capacity. Therefore, it is imperative to reduce oxidative damage and enhance the antioxidant capacity of the organism. In the current study, there was a significant increase in T-AOC and SOD enzyme activities in the S3 and S3B9 groups, an increase in MDA and GSH-Px enzyme activities in the S3 group, and a significant increase in POD enzyme activity in the S3B9 group. These results align with the findings of Chen et al. [60], who observed a significant increase in GSH-Px and SOD activities in the hepatopancreas of shrimp fed with *Rhodotorula mucilaginosa* and *B. licheniformis*. Ghasem et al. [61] demonstrated that the activities of GSH-Px and SOD in Nile tilapia were significantly enhanced following the addition of PHDP and *Pediococcus acidilactici* to their diet. Similarly, feeding *Gracilaria tenuistipitata* extract to *L. vannamei* resulted in an increase in SOD activity [62]. Furthermore, the supplementation of dietary with 1 g/kg of seaweed polysaccharides led to elevated levels of T-AOC, SOD, and GSH-Px in the hemolymph of *Fenneropenaeus merguiensis* [63]. The observed phenomenon can be ascribed to the abundance of active biological compounds such as polyphenols, glycosides, anthocyanins, tannins, and thiocarbamates in seaweed. These compounds contribute to the high levels of natural antioxidants that effectively scavenge free radicals, stimulate the production of active antioxidant enzymes, and inhibit oxidative enzymes, thereby mitigating the detrimental effects of free radicals [64]. Additionally, the synthesis of extracellular polysaccharides by prebiotic bacteria plays a role in inhibiting ROS and modulating antioxidant-related genes, ultimately enhancing the antioxidant capacity of the system [65]. The antioxidant capabilities of algae have been evidenced in prior research through a variety of mechanisms, including the reduction of hydrogen peroxide radicals and their conversion into oxygen and water [66]. Furthermore, the observed trends in SOD and GSH-Px activities in the hepatopancreas of shrimp were found to align with their respective mRNA levels.

The Inclusion of 3% *G. lichenoides* in the feed resulted in a significant upregulation of *mtMnSOD*, *GSH-Px*, *Trx*, *Hippo*, and *Nrf2* genes. Furthermore, the addition of 3% *G. lichenoides* in combination with *B. amyloliquefaciens* (10^9^ CFU/g) led to an increase in the relative expression levels of *CAT*, *GSH-Px*, *Trx*, and *Hippo* genes. This study further examined antioxidant-related genes’ expression levels of hepatopancreas in shrimp supplemented with additives, based on the analysis of physiological and biochemical indices. Under conditions of oxidative stress, *Nrf2* levels are maintained through a reduction in ubiquitination and degradation, as well as activation of downstream antioxidant factors [67]. *Nrf2* is regulated by multiple pathways at various stages, forming a dimer with sMAF proteins and controlling the expression of genes containing antioxidant response elements (ARE), such as *GPx*, *Trx*, *Prx*, *TrxR*, *CAT*, and *SOD* [68]. This process mediates the activation of a diverse array of antioxidant genes through the Keap1-Nrf2- ARE signaling pathway [67,69]. The *SOD* and *CAT* genes in zebrafish was found to be significantly increased by addition of *G. gracilis* [70]. Furthermore, Chen et al. [59] conducted a study which demonstrated that the inclusion of mannan oligosaccharide (MOS) and *B. lincheniformis* in the diet resulted in a significant upregulation of the expression levels of *CAT*, *GPx*, and *SOD* in *L. vannamei*. The inclusion of 3% *G. lichenoides* in the diet led to an upregulation of *Nrf2*, *SOD*, and *GSH-Px* gene expression, aligning with the observed antioxidant enzyme activity levels. These findings indicate that even a modest quantity of algal meal can enhance antioxidant enzymes through the stimulation of gene expression. Curcumin was observed to enhance the activities of antioxidant enzyme SOD, CAT, and GPx by upregulating the expression of *Cu/Zn SOD*, *MnSOD*, *CAT*, and *GPx* mRNA, which was associated with the *Keap1/Nrf2* signaling pathway in grass carp following infection with *Aeromonas hydrophila* [59]. *Keap1* is capable of regulating the expression of antioxidant genes, such as *CAT* and *GSH-Px*, through modulation of the *Nrf2* signaling pathway. Moreove, a fraction of *Keap1* may potentially increase the efficacy of hepatopancreatic antioxidant enzymes by activating genes through the *Nrf2* signaling pathway [69]. The findings indicated elevated levels of *Trx* expression in the S3, S3B9, and S8B11 groups, aligning with previous research demonstrating the ability of aspalathin to mitigate lead-induced renal oxidative stress in mice through modulation of the *Trx* and *Nrf2* signaling pathways [71]. This mechanism involves the binding of Nrf2 to the antioxidant ARE and subsequent upregulation of *Trx* and *TrxR* genes via the *Trx* signaling pathway, ultimately reducing oxidative stress-induced damage [71]. The inclusion of 8% *G. lichenoides* in combination with *B. amyloliquefaciens* (10^11^ CFU/g) in the diet resulted in an upregulation of *Nrf2*, *CAT*, and *Trx* gene levels. However, there was no notable impact on antioxidant enzyme activities, indicating the necessity for further exploration into the specific mechanisms underlying this effect. The study revealed that the supplementation of *B. amyloliquefaciens* (10^9^ CFU/g) in the diet did not have significant alterations in the antioxidant enzyme activities and expression levels of relevant antioxidant genes in *P. monodon*. Conversely, the concurrent administration of modest quantities of algal meal and *B. amyloliquefaciens* induced the activation of the antioxidant response factor *Nrf2*, thereby facilitating the expression of diverse antioxidant enzymes and enhancing the transcription of downstream genes including *Trx*, *SOD*, *CAT*, and *GPx*. It is hypothesized that the coexistence of low levels of *G. lichenoides* and *B. amyloliquefaciens* may result in a heightened presence of natural antioxidants.

In the process of nutrient assimilation, the intestines are essential due to the increased surface area for absorption provided by the folds within them [14]. In the present study, the experimental group demonstrated a greater abundance of intestinal tissue epithelia with well-defined brush borders and densely packed nuclei compared to the control group. Additionally, the study found that the height of intestinal epithelial cells in *P. monodon* fed a diet containing *G. lichenoides* was significantly greater than those in the S8B11 group, indicating that the inclusion of *G. lichenoides* in the diet may enhance the nutrient absorption capacity of the shrimp. Previous research has shown that seaweed (*Enteromorpha*) can increase villus width and lead to a larger villus surface area [63]. Paul et al. [72] illustrated that the inclusion of dietary *K. alvarezii* resulted in a notable enhancement of villus width and crypt depth within the intestine of broiler chicken. The supplementation of seaweeds in cultured animals has been shown to augment intestinal microvilli and increase absorptive surface area, thereby potentially enhancing nutrient utilization and promoting growth. In the current study, the highest values of intestine epithelium in *P. monodon* were observed in the S3 group. Additionally, elevated values of fold depth were noted in the S3B9, B9, and B11 groups, which were significantly greater than those in the control group. Previous research has shown that *B. subtilis* can enhance broiler villus height, villus surface area, and absorbing epithelial cell area [53], while the supplementation of *B. licheniformis* has been found to promote the development of intestinal villi in common carp, with well-developed and well-arranged villi [73]. These findings suggest that *Bacillus* have the potential to improve the structure integrity of intestinal tissue. 

The intestinal microbiota serves as the primary locus for organ digestion and absorption, supplying essential nutrients and energy to the host organism while also exerting a significant influence on the growth, health, and developmental processes of shrimp [38]. In this study, there were no significant differences in the diversity and richness of microbial communities. Diversity and richness are comprehensive indicators composed of a large number of bacteria, indicating that the type of feed added does not change the total type and number of intestinal florae [38]. But there are differences in microbial community composition: at the phylum level, Proteobacteria, Bacteroidetes, Actinobacteria, and Fusobacteria were identified as the predominant bacterial taxa in the intestinal microbiota of *P. monodon*. Previous studies have shown that Proteobacteria, whose microbiota is dominated by Gram-negative bacteria, have a high risk of disease and reduce the health of the host [74]. In the study by Wang et al. [75], the microbial community of Proteobacteria in the intestine of *L. vannabaeus* was increased after white spot syndrome virus (WSSV) infection. Actinobacteria have been shown to degrade organic matter, synthesize antibacterial compounds [76], and modulate inflammatory and autoimmune responses through the induction of regulatory T cells, potentially enhancing disease resistance and immunity in shrimp [77,78]. Firmicutes and Bacteroidetes are generally the major microbiota in the gut of humans and other mammals, participating in the digestion and metabolism of food by the host [79], and are generally thought to contribute to improved digestion in aquatic animals [80]. The relative abundance of Firmicutes in the S3 group was significantly higher than that in the S3B9, S8, and B11 groups. Firmicutes is a commonly predominant microbial taxa in the gastrointestinal tract of humans and other mammals, playing a significant role in food digestion and metabolism within the host organism [73,81]. In aquatic animals, these microbial groups are believed to enhance digestive processes. Firmicutes have been shown to facilitate the absorption of fatty acids in host organisms and promote the synthesis of short-chain fatty acids [79], a process closely associated with the development of fatty liver [38]. In light of the alterations observed in the levels of FBW, WGR, SGR, and intestinal lipase activity within the S3 group, the findings indicate that the supplementation of *G. lichenoides* at a low concentration may enhance digestive enzyme activities and growth performance through modulation of the Firmicutes composition. At the taxonomic level of genus, we identified several dominant genera were identified that displayed significant differences among the groups, including *Vibrio*, *Shimia*, and *Lactobacillus*. *Vibrio*, a facultative pathogen, exhibited high enrichment in the intestinal microbiota of aquatic animals [82]. Our findings indicated a significantly relative abundance of *Vibrio* in the S3 group compared to the S3B9. Previous research by Duan et al. [83] supported the enrichment of *Vibrio* in the intestinal microbiota of the fast-growing shrimp, aligning with the results of our study. The growth performance of shrimp in the S3 group was significantly highest.

## 5. Conclusions

The inclusion of dietary *G. lichenoides* and *B*. *amyloliquefaciens* positively impacted the growth, antioxidant capacity, and microbial composition of shrimp. Adding 3% *G. lichenoide* had significantly higher FBW, WGR, and SGR; the antioxidant capacity in hepatopancreatic tissues of *P. monodon* was significantly increased in the 3% *G. lichenoide* group and the 3% *G. lichenoide* + 10^9^ CFU/g *B. amyloliquefaciens* group; antioxidant gene expression was significantly increased in the hepatopancreas of shrimp between the 3% *G. lichenoide* group and the 3% *G. lichenoide* + 10^9^ CFU/g *B. amyloliquefaciens* group; supplementing the diet with the *G. lichenoide* and *B. amyloliquefacients* improved the stomach and intestinal lipase activity of *P. monodon*; supplementing the diet with *B. amyloliquefaciens* (10^9^ CFU/g) improved the intestinal tissue structure of *P. monodon*; adding *B. amyloliquefaciens* (10^9^ CFU/g) and 8% *G. lichenoide* + *B. amyloliquefaciens* (10^11^ CFU/g) had a significant impact on the expression of genes involved in shrimp intestinal lipid metabolism; and the relative abundance of intestinal microbiome in the control, S3, B11, S3B9, and S8B11 groups had a significant impact. The results indicate that the inclusion of 3% *G. lichenoides* led to specific improvements in shrimp. This research investigated the impacts of *G. lichenoide* and *B. amyloliquefaciens* on growth, antioxidant capacity, and intestinal microbiome, offering a theoretical foundation for further study in this area.

## Figures and Tables

**Figure 1 biology-13-00252-f001:**
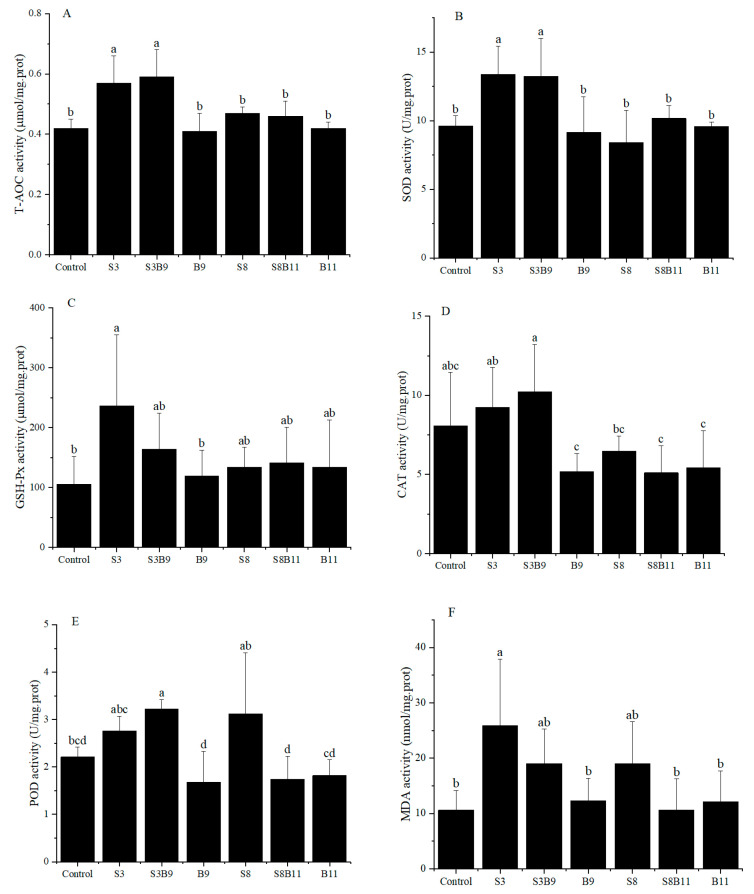
(**A**) T-AOC, (**B**) SOD, (**C**) CAT, (**D**) GSH-Px, (**E**) POD, and (**F**) MDA content in the hepatopancreas of shrimp fed different diets for 56 days. Vertical bars represented the mean ± SD (*n* = 4) and data indicated with different letters were significantly different (*p* < 0.05) among treatments, the same as below.

**Figure 2 biology-13-00252-f002:**
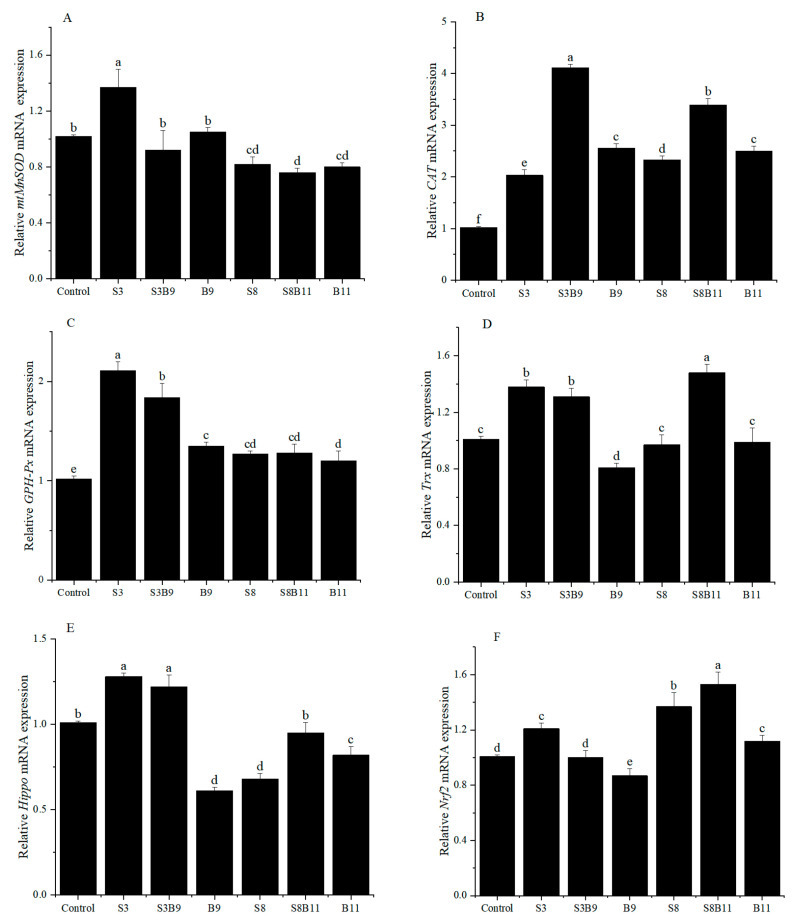
Expression levels of antioxidant genes in the hepatopancreas of shrimp fed different diets for 56 days. (**A**) *mtMnSOD*, (**B**) *CAT*, (**C**) *GSH-Px*, (**D**) *Trx*, (**E**) *Hippo*, (**F**) *Nrf2*. The reference gene is *EF*-1α. Vertical bars represented the mean ± SD (*n* = 4) and data indicated with different letters were sig-nificantly different (*p* < 0.05) among treatments.

**Figure 3 biology-13-00252-f003:**
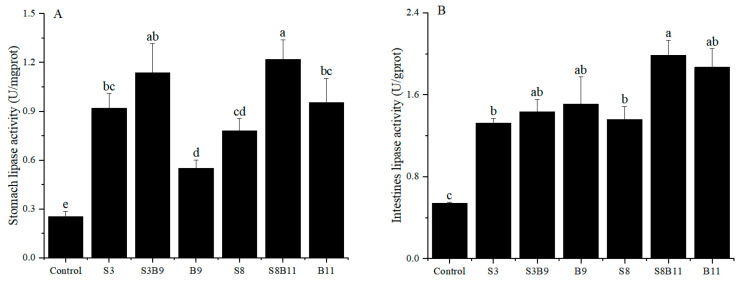
Lipase activity in the stomach and intestine of shrimp fed different diets for 56 days. (**A**) Lipase activity of *P. monodon* in the stomach. (**B**) Lipase activity of *P. monodon* in the intestine. Vertical bars represented the mean ± SD (*n* = 4) and data indicated with different letters were significantly different (*p* < 0.05) among treatments.

**Figure 4 biology-13-00252-f004:**
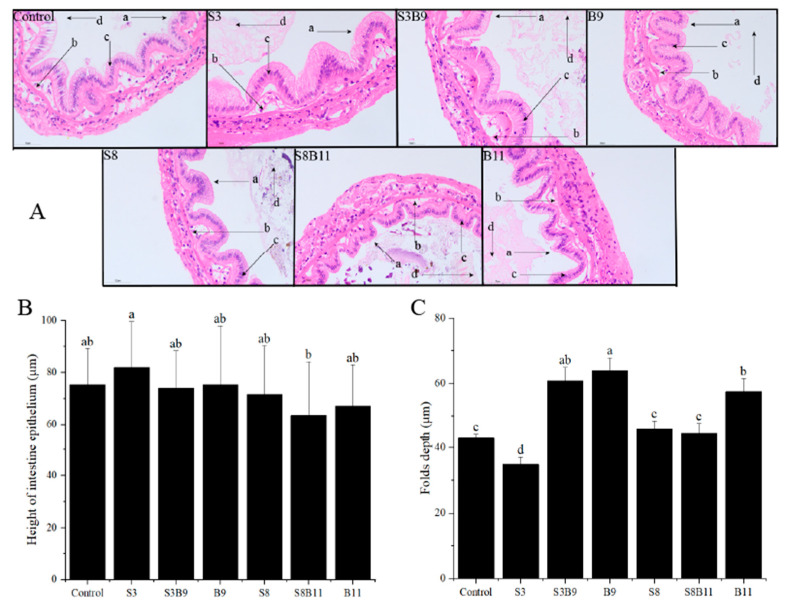
Intestinal structure of *P. monodon* fed the different experimental diets for 56 days. (**A**) Intestine sections of shrimp fed different diets, ×400. (**B**) Height of intestine epithelium. (**C**) Folds depth of intestine in shrimp fed different diets. The letters in figure (**A**) indicate: (a) brush border, (b) epithelium, (c) nuclei, (d) lumen. Vertical bars represented the mean ± SD (*n* = 4) and data indicated with different letters were significantly different (*p* < 0.05) among treatments.

**Figure 5 biology-13-00252-f005:**
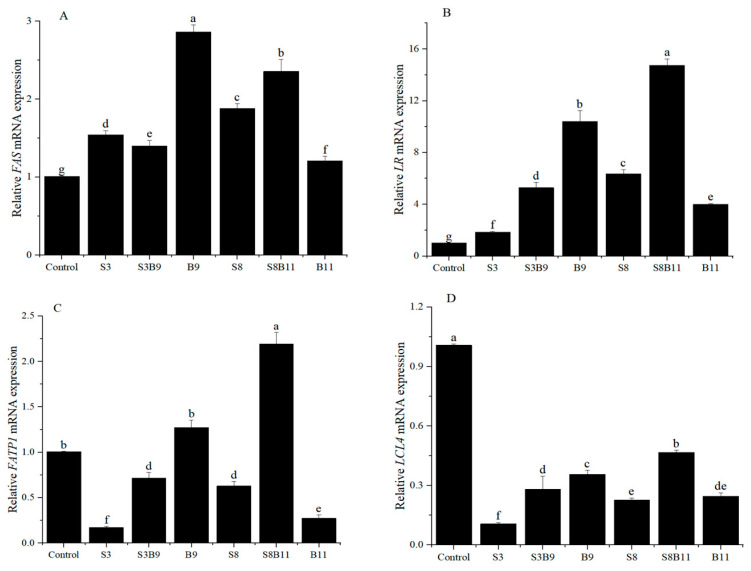
*FAS* (**A**), *LR* (**B**), *FATP*1 (**C**), and *LCL*4 (**D**) gene expression level in the intestine of shrimp fed different diets for 56 days. The reference gene is *EF*-1*α*. Vertical bars represented the mean ± SD (*n* = 4) and data indicated with different letters were significantly different (*p* < 0.05) among treatments.

**Figure 6 biology-13-00252-f006:**
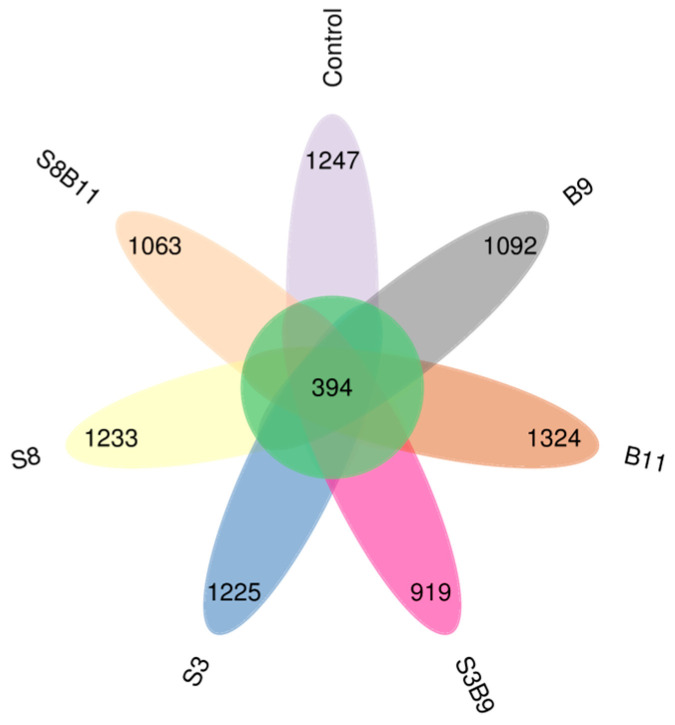
Shared OUTs Venn analysis.

**Figure 7 biology-13-00252-f007:**
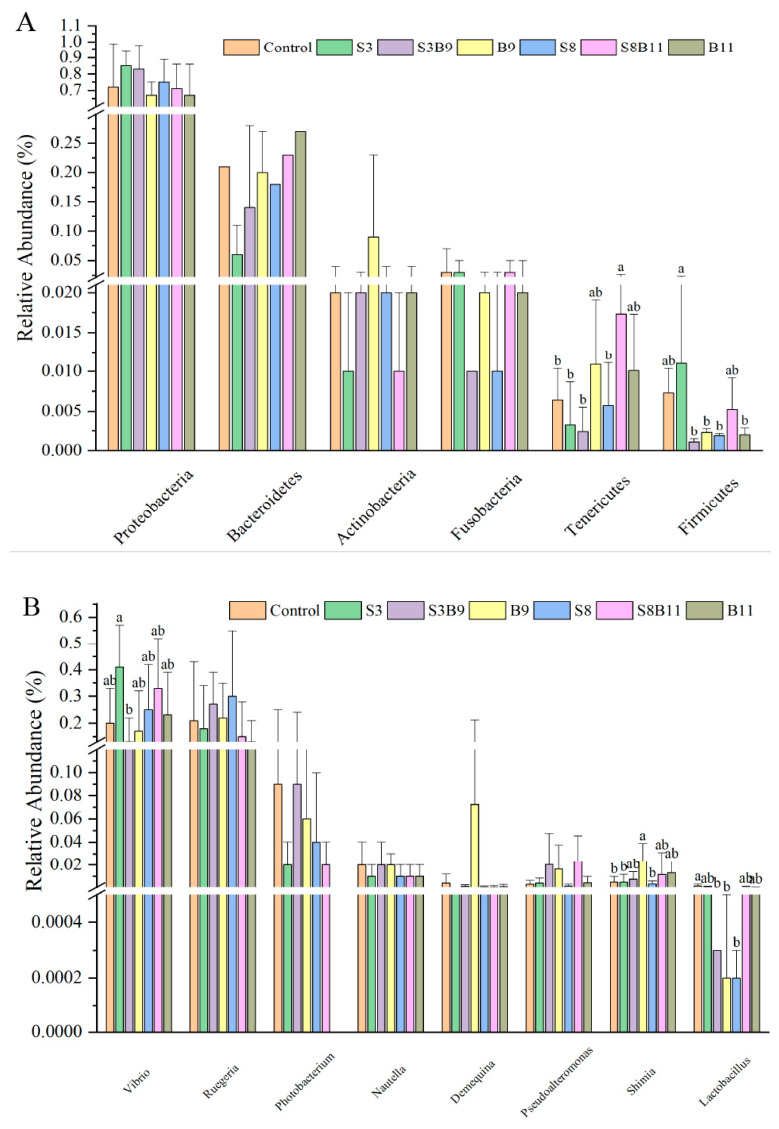
Intestine microbe composition of *P. monodon* that were fed the different experimental diets for 56 days. (**A**) Relative abundance of major bacterial phyla in *P. monodon* fed with the different experimental diets for 56 days. (**B**) Relative abundance of major bacterial genera in *P. monodon* fed with the different experimental diets for 56 days. (**C**) Heat map analysis of intestine microbial on the top 50 genera. Vertical bars represented the mean ± SD (*n* = 4) and data indicated with different letters were significantly different (*p* < 0.05) among treatments.

**Figure 8 biology-13-00252-f008:**
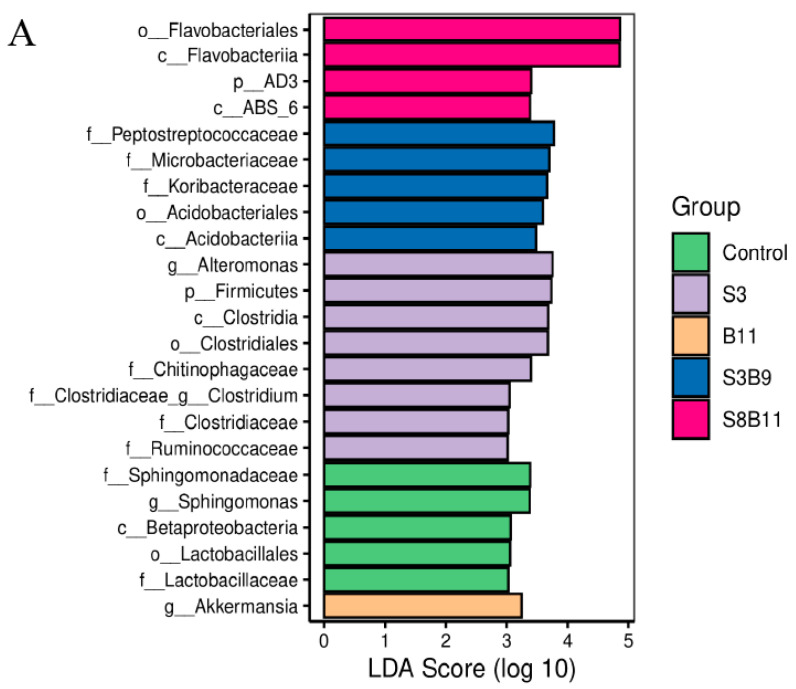
Indicator species analysis of intestinal microbiota of *P. monodon* fed the different experimental diets for 56 days. (**A**) LDA score of Lefse-PICRUSt. The length of the column represents the effect size of bacterial lineages. (**B**) Lefse cladogram. The cladogram was obtained by mapping differences onto a known hierarchical classification tree. Green: bacterial taxa enriched in the control; Purple: bacterial taxa enriched in the S3 group; Orange: bacterial taxa enriched in the B11 group; Blue: bacterial taxa enriched in the S3B9 group; Pink: bacterial taxa enriched in the S8B11; White: no significant differences. Only the taxa that had a linear discriminate analysis (LDA) value above 2.0 are shown.

**Table 1 biology-13-00252-t001:** Composition and nutrient contents of experimental diets (%).

Items (%)	Control	S3	S3B9	B9	S8	S8B11	B11
Fish meal	31.00	31.00	31.00	31.00	31.00	31.00	31.00
Soybean meal	17.00	17.00	17.00	17.00	17.00	17.00	17.00
Peanut meal	15.00	15.00	15.00	15.00	15.00	15.00	15.00
Wheat flour	20.30	20.00	19.90	20.29	19.50	19.49	20.29
*Gracilaria lichenoides*	0.00	0.30	0.30	0.00	0.80	0.80	0.00
*Bacillus amyloliquefaciens*	0.00	0.00	0.01	0.01	0.00	0.01	0.01
Beer yeast	5.00	5.00	5.00	5.00	5.00	5.00	5.00
Krill meal	5.00	5.00	5.00	5.00	5.00	5.00	5.00
Soy lecithin	1.00	1.00	1.00	1.00	1.00	1.00	1.00
Fish oil	1.00	1.00	1.00	1.00	1.00	1.00	1.00
Soybean oil	1.00	1.00	1.00	1.00	1.00	1.00	1.00
Choline chloride (50%)	0.60	0.60	0.60	0.60	0.60	0.60	0.60
Monocalcium phosphate	1.00	1.00	1.00	1.00	1.00	1.00	1.00
Vitamin premix ^a^	1.00	1.00	1.00	1.00	1.00	1.00	1.00
Mineral premix ^b^	1.00	1.00	1.00	1.00	1.00	1.00	1.00
Ascorbic Phosphate ester	0.10	0.10	0.10	0.10	0.10	0.10	0.10
Sum	100.00	100.00	100.00	100.00	100.00	100.00	100.00
Nutrient levels ^c^							
Moisture	9.66	9.97	10.17	9.46	9.86	10.32	9.40
Ash	10.35	11.29	11.53	11.28	13.18	13.64	11.27
Crude protein	40.62	40.73	43.87	41.78	40.15	40.44	39.71
Crude lipid	9.12	8.95	8.73	9.09	9.32	8.99	8.84
Crude fiber	3.94	3.29	4.30	4.52	3.21	3.64	3.73

Note: ^a^. Purchased from Guangzhou Feixite Aquatic Technology Co., Ltd (Guangzhou, China).Vitamin premix (g kg^−1^): VE, 75; VK, 2.5; VB_1_, 0.25; VB_2_, 1.0; VB_3_, 5.0; VB6, 0.75; VB_12_, 2.5; VA, 2.5; VD, 6.25; folic acid, 0.25; cellulose, 500; inositol, 379; biotin, 2.5. ^b^. Purchased from Guangzhou Xinhai Lisheng Biotechnology Co., Ltd (Guangzhou, China). Mineral premix (g kg^−1^): KCl, 90; NaCl, 40; KI, 0.04; ZnSO_4_·7H_2_O, 4; CuSO_4_·5H_2_O, 3; CoSO_4_·7H_2_O, 0.02; MnSO_4_·H_2_O, 3; FeSO_4_·7H_2_O, 20; MgSO_4_·7H_2_O, 124; CaCO_3_, 215; Ca(H_2_PO_4_) _2_·2H_2_O, 500. ^c^. Nutrient levels in dry-weight feed.

**Table 2 biology-13-00252-t002:** The primers for real-time fluorescence quantification PCR.

cDNA	Forward Primer (5′–3′)	Reverse Primer (5′–3′)	Size of Production	GenBank Nos.
*mtMnSOD*	TCGCCGCCAGGAGACTCTTC	GGCACAGATGACAGGTTCCAAGG	122	KC461130.1
*CAT*	GTCCTTCTTCAGCAGCCTCAGTTG	CTTGGCTCGTGGTCAGGTTATCG	155	KR908786.1
*GSH-Px*	CGTCCGTCCTGGCAATAACTTCG	CTGGCAGCGGCAGTCGTTC	116	JX912159.1
*Trx*	CCTGAAGGTGGATGTGGACGAATG	AGTGGAATGGATTACTTGTGCTTCTCG	168	JF828310.1
*Hippo*	TGAGCACAACCAAACCCACCATC	CATCGTCCGACTGTCCACTTCATC	88	Not acquired
*Nrf2*	CCAACCTCCAGTAACAAGCCAAGAG	TCAACAATTCTGATGAGCACAGCAATG	237	MW390830.1
*FAS*	GCGTGATAACTGGGTGTCCT	CTTTCAGGCCCTGGATGATA	227	XM030250746.2
*LR*	GTGGCAGGAATGAGTGTGAGG	GCATACCCTCCAACGCACTC	215	NM009309.2
*FATP1*	GATGACACGGACATGACTACTG	CTCCTGGCTTCAGCACATTCC	195	NM164934.2
*LCL4*	CTCCAATCTGTAGTTTAACCAAGTCC	GATGACACGGACATGACTACTG	304	XM006583136.4
*EF*-1α	AGTATGCTCCTTTTGGACGTTTTGC	CCTTTTCTGCGGCCTTGGTAGTC	120	GU136229.1

Note: *Nrf2*, NF-E2-related factor 2; *Trx*, thioredoxin; *FAS*, fatty acid synthetase; *LR*, lipophorin receptor; *FATP*1, fatty acid transport protein 1; *LCL*4, long-chain fatty acid-CoA ligase 4; *EF*-1α, elongation factor 1α.

**Table 3 biology-13-00252-t003:** Effects of different diet treatments on growth performance of shrimp after feeding for 56 days (*n =* 4).

Items	IBW (g)	FBW (g)	WGR (%)	SGR (%)	FCR	SR (%)
Control	1.06 ± 0.02	9.18 ± 0.48 ^b^	765.56 ± 56.72 ^b^	3.92 ± 0.12 ^b^	1.79 ± 0.17	90.00 ± 6.67
S3	1.04 ± 0.06	10.49 ± 0.90 ^a^	908.94 ± 33.58 ^a^	4.20 ± 0.06 ^a^	1.50 ± 0.24	90.00 ± 0.00
S3B9	1.05 ± 0.05	9.47 ± 0.62 ^ab^	804.95 ± 56.43 ^b^	4.00 ± 0.12 ^ab^	1.64 ± 0.14	94.44 ± 1.93
B9	1.04 ± 0.02	8.95 ± 0.56 ^b^	759.42 ± 41.23 ^b^	3.91 ± 0.09 ^b^	1.53 ± 0.30	88.89 ± 3.85
S8	1.02 ± 0.03	8.51 ± 0.45 ^b^	731.19 ± 49.05 ^b^	3.85 ± 0.11 ^b^	1.80 ± 0.10	93.33 ± 0.00
S8B11	1.01 ± 0.04	9.08 ± 1.05 ^b^	794.15 ± 80.75 ^b^	3.98 ± 0.17 ^b^	1.46 ± 0.12	93.33 ± 0.00
B11	1.04 ± 0.04	8.33 ± 0.29 ^b^	703.55 ± 17.80 ^b^	3.79 ± 0.04 ^b^	1.63 ± 0.21	93.33 ± 8.82

Note: Values with different letters in the same column indicate significant differences (*p* < 0.05).

**Table 4 biology-13-00252-t004:** Body composition of shrimp.

Items	Control	S3	S3B9	B9	S8	S8B11	B11
Moisture (%)	76.51 ± 1.20	77.02 ± 0.45	76.15 ± 0.86	76.76 ± 0.17	76.54 ± 0.47	76.09 ± 0.81	76.00 ± 0.74
Protein (% dry matter)	70.01 ± 1.30	70.37 ± 0.68	70.45 ± 0.35	70.56 ± 1.27	71.09 ± 0.42	71.39 ± 0.87	70.01 ± 0.94
Lipid (% dry matter)	10.40 ± 0.70 ^ab^	10.49 ± 1.11 ^ab^	10.73 ± 1.86 ^ab^	11.08 ± 0.75 ^a^	9.80 ± 1.06 ^ab^	9.35 ± 0.92 ^ab^	9.27 ± 0.76 ^b^
Ash (% dry matter)	16.61 ± 0.69 ^ab^	16.11 ± 1.13 ^ab^	15.67 ± 0.48 ^b^	16.97 ± 0.56 ^a^	16.66 ± 0.72 ^ab^	16.11 ± 0.79 ^ab^	16.46 ± 0.55 ^ab^

Note: Values in the same row with different letters are significantly difference (*p* < 0.05).

**Table 5 biology-13-00252-t005:** Intestinal microbial diversity of *P. monodon* fed the different experimental diets for 56 days.

Alpha name	Control	S3	S3B9	B9	S8	S8B11	B11
Chao1	1066.90 ± 445.03	1054.39 ± 354.66	876.95 ± 243.56	1139.16 ± 399.85	1229.12 ± 352.32	1052.77 ± 469.62	1298.55 ± 404.43
Simpson	0.94 ± 0.03	0.91 ± 0.05	0.91 ± 0.04	0.94 ± 0.03	0.95 ± 0.02	0.92 ± 0.04	0.93 ± 0.01
Shannon	5.74 ± 0.97	5.39 ± 0.41	5.14 ± 0.94	5.78 ± 0.83	5.94 ± 0.46	5.20 ± 0.97	5.54 ± 0.57
Coverage (%)	99.67 ± 0.17	99.65 ± 0.14	99.73 ± 0.05	99.59 ± 0.15	99.61 ± 0.15	99.61 ± 0.19	99.54 ± 0.13

## Data Availability

Data are contained within the article.

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
