# Peer review of "Effects of Dietary Gracilaria lichenoides and Bacillus amyloliquefaciens on Growth Performance, Antioxidant Capacity, and Intestinal Health of Penaeus monodon"

_biology, 2024, doi:10.3390/biology13040252_

Round 1

Reviewer 1 Report

Comments and Suggestions for Authors

The papers are well balanced with technical and scientific data on the effect of using G. lichenoides and B. amyloliquefaciens diet in feed on the growth, antioxidant capacity, and microbial composition of shrimp. This paper provides important knowledge about alternative methods to eliminate the effects of antibiotic use in diseases of aquatic animals.

Author Response

Abstract

  1. The abstract is well written, but I suggest summarizing theexperimental data presented.

Answer:  I added the experimental data in abstract. Please check them in line 38-39. 

Introduction

  1. Line106- 107: I recommend that Latin names be written with slanted text, e.g. Bacillus.

Answer:  I have rewritten the Latin names with slanted text. Please check it in Line128.

  1. Line109: The expression " As far as we know, " I recommend to be reviewed and expressed

Answer: I have deleted the expression “As far as we know” in Line 137.

  1. Irecommend improving the introduction chapter with recent and relevant concerns and research.

Answer: I have improved the introduction chapter with recent and relevant concerns and research. Please check them in Line 91- 106, Line 108- 113, Line 130- 136.

Materials and methods

  1. Materialsand methods are suggestively designed and  I recommend formatting to justify the material and methods chapter.

Answer : I have revised the materials and methods chapter and added specific operation methods. Please check them in the chanpter of materials and methods.

  1. The   methodology   used   involvesthe   use    of   two   different    concentrations   of    amyloliquefacients of 109 CFU/g and 1011  CFU/g in the diet. It would be recommended to use the same concentrations.

Answer: Thank you very much for your advice. In this study we mainly want to investigate the effects of high or low levels of G. lichenoides and B. amyloliquefaciens alone or the combination to P. mondon. Your suggestion is of great research value. In the following study, our research team will set the same concentration of G. lichenoides or B. amyloliquefaciens, change the concentration of another additive, study the influence on P. mondon, compare the effect, and find the optimal dosage of additive.

Line 129: I recommend that Latin names be written with slanted text in the whole manuscript, e.g. B. amyloliquefacients, table 1. Composition and nutrient contents of experimental diets. In Table 1 please insert the units of measurement of each diet component used.

Answer : Thank you for your correction. I have italicized the species names in Table 1. The format of the official website template table title used in this article has been modified. The units of measurement of each diet component used is % at line 172.

Line 184: You stated that lipase activity in the stomach and intestine has been assayed with commercial assay kits, please list at least 5 scientific articles that have used such kits for these assays mentioned above, validating the results obtained comparatively.

Answer: The 5 scientific articles about the lipase activity assaying with commercial assay kits were showed as follows.

  1.  Duan, Y.; Zhang, J.; Huang, J.; Jiang, S. Effects of dietary Clostridium butyricumon the growth, digestive enzyme activity, antioxidant capacity, and resistance to nitrite stress of Penaeus monodon. Probiotics & Antimicro. Prot.2019, 11, 938–945.
  2.  Wang, Y., Li, Z., Li, J., Duan, Y. F., Niu, J., Wang, J., Huang, Z., Lin, H. Z. Effects of dietary chlorogenic acid on growth performance, antioxidant capacity of white shrimp Litopenaeus vannameiunder normal condition and combined stress of low-salinity and nitrite. Fish & shellfish immunology, 2015. 43(2), 337-345.
  3. 3.  Ming, J.; Ye, J.; Zhang, Y.; Yang, X.; Shao, X.; Qiang, J.; Xu, P. Dietary optimal reduced glutathione improves innate immunity, oxidative stress resistance and detoxification function of grass carp (Ctenopharyngodon Idella) against Microcystin-LR. Aquaculture 2019, 498, 594–605.
  4.  Li, R.; Zhou, Y.; Ji, J.; Wang, L. Oxidative damages by cadmium and the protective effects of low-molecular-weight chitosan in the freshwater crab (Sinopotamon yangtsekienseBott 1967): Oxidative damages by Cd and the protective effects of LMWC. Aquaculture Research 2011, 42, 506–515.
  5.  Yu, X.; Wang, X.-P.; Lei, F.; Jiang, J.-F.; Li, J.; Xing, D.-M.; Du, L.-J. Pomegranate leaf attenuates lipid absorption in the small intestine in hyperlipidemic mice by inhibiting lipase activity. Chinese Journal of Natural Medicines2017, 15, 732–739.

Discussion

The discussions are well formulated and comprehensive in terms of the subject discussed, but can be improved especially regarding microbial community composition, where the results were remarkable.

Answer: In the discussion chapter, I have added a section about microbial community composition. Please check them in line 634-638 and line 640-650.

Conclusions

Conclusions are adequate, but not sufficiently elaborated, I recommend highlighting the relevance  of the  study.  Conclusions  should  be  adequately  substantiated  to  achieve  the proposed working hypothesis and specify which completed studies will be developed in future studies. Please explain in the conclusions the extent to which the aim and objective of the study have been achieved.

Answer: Thank you for your advice. We fully agree with your suggestion. In the Conclusions, we have summarized the experimental results in detail. Our research provides a theoretical basis for researchers to study the effects of seaweed and probiotics on shrimp. In the next part, more theoretical basis can be provided to study the effects of different types of seaweed and probiotics on aquatic animals. We have added the aim and objective of the study in the manuscript. Please check in line 672-688.

References

The references are adequate but can be improved with recent studies.

Answer : Thanks for your suggestion. I have added recent references in line 743-763, line 773-778, line 791-802 and line 896-911.

Reviewer 2 Report

Comments and Suggestions for Authors

The work has an innovative theme and the analysis corroborates the results, I really liked the writing and the robustness of the work. I have a few corrections and I believe that the work can be published and will serve as a knowledge base for many others to come. 

1. I believe there is a missing hypothesis for the work at the end of the introduction

2. Describe the rate of feeding and how the food was offered, even if there is a reference.

3. Describe the system used to raise the animals, e.g. what was the aeration system like? How did the air get into the tanks? Porous stone, hoses?

4. How soon were the food pellets considered to be leftovers removed from the tank to correct feed consumption? I say this because the length of time the feed remains in the tank is extremely important, both for the animal's own nutrition and so that the feed doesn't leach out, since you used a temperature of 29 degrees Celsius during the experiment.

5. I think the authors could better organize the description of the animals' cultivation. I can understand it, but I had to go back or forward to clear up some doubts. I think that, for a fluid and dynamic reading, this text could be improved and details could be given.

6. In Taleba 3 and in the rest of the text, when necessary, describe what the acronyms used mean.

Author Response

Response letter 2

  1. I believe there is a missing hypothesis for the work at the end of the introduction

Answer: I added the predicted effects in the last paragraph of the introduction, at line 140-148. 

  1. Describe the rate of feeding and how the food was offered, even if there is a reference.

Answer: I added the rate of feeding and how the food was offered in line 185-191.

  1. Describe the system used to raise the animals, e.g. what was the aeration system like? How did the air get into the tanks? Porous stone, hoses?

Answer: I wrote the part of the culture system in detail, including how to feed, how to change water, and the aeration system, and oxygen is connected to the oxygen supply system by a hose air stone, which is filled day and night, at line 185-197. 

  1. How soon were the food pellets considered to be leftovers removed from the tank to correct feed consumption?I say this because the length of time the feed remains in the tank is extremely important, both for the animal's own nutrition and so that the feed doesn't leach out, since you used a temperature of 29 degrees Celsius during the experiment.

Answer: After feeding for 1 hours later, the leftovers in the bait dish was collected, rinsed 2 to 3 times with water, dried at 60° C and weighed. I added the food collection at line 190-192.

  1. I think the authors could better organize the description of the animals' cultivation. I can understand it, but I had to go back or forward to clear up some doubts. I think that, for a fluid and dynamic reading, this text could be improved and details could be given.

Answer: I carefully improved the description of the shrimps' cultivation in Line 188-195 and improved some details.

  1. In Taleba 3 and in the rest of the text, when necessary, describe what the acronyms used mean.

Answer: I added the relevant content in the Note under the Table 2 and Table 3, at line 237-239 and line 273.